# Statistical Triage Model for Feline Infectious Diseases in a Veterinary Isolation Unit: The Case of Feline Immunodeficiency and Leukemia Viruses

**DOI:** 10.3390/vetsci12090902

**Published:** 2025-09-17

**Authors:** Miguel M. Maximino, Inês C. Machado, Telmo P. Nunes, Luís M. Tavares, Virgílio S. Almeida, Solange A. Gil, Nuno Sepúlveda

**Affiliations:** 1Faculty of Veterinary Medicine, University of Lisbon, Av. Universidade Técnica, 1300-477 Lisbon, Portugal; miguelmendesmaximinoo@gmail.com (M.M.M.); inesmachado@fmv.ulisboa.pt (I.C.M.); tnunes@fmv.ulisboa.pt (T.P.N.); ltavares@fmv.ulisboa.pt (L.M.T.); vsa@fmv.ulisboa.pt (V.S.A.); solange@fmv.ulisboa.pt (S.A.G.); 2CIISA—Centre of Interdisciplinary Research in Animal Health, Faculty of Veterinary Medicine, University of Lisbon, Av. Universidade Técnica, 1300-477 Lisbon, Portugal; 3Associate Laboratory for Animal and Veterinary Sciences (AL4AnimalS), 1300-477 Lisbon, Portugal; 4Teaching Hospital, Faculty of Veterinary Medicine, University of Lisbon, Av. Universidade Técnica, 1300-477 Lisbon, Portugal; 5Faculty of Mathematics and Information Science, Warsaw University of Technology, Koszykowa 75, 00-637 Warsaw, Poland; 6CEAUL—Centro de Estatística e Aplicações da Universidade de Lisboa, 1749-016 Lisbon, Portugal

**Keywords:** cats, Feline Immunodeficiency Virus, Feline Leukemia Virus, infectious diseases, predictive models, triage

## Abstract

In cats, as in humans, the success of a given treatment depends on a timely diagnosis and its accuracy. This is particularly important for making decisions on suspected cases referred to isolation units dedicated to diagnosing, treating, and containing feline infectious diseases. In practice, the actions of these units are bounded by their available resources, making triage difficult to perform rapidly without increasing the cost. In this study, we explored the possibility of constructing simple and rapid statistical models for triaging suspected cases of two feline viral infections using routinely collected data (e.g., age, sex, and neutered status) from a Portuguese Biological Isolation and Containment Unit. Our study focused on the serious but common infections caused by Feline Immunodeficiency and Leukemia viruses. The constructed triage models based on these simple data had moderate sensitivity and specificity. This finding suggested that these models should be constructed with additional triage factors before being deployed to the real-world clinical practice.

## 1. Introduction

Managing infectious diseases in veterinary hospitals poses challenges due to the diversity of species treated and the dynamic microbial environment they create. Close contact between animals, staff, and shared clinical spaces increases the risk of pathogen transmission and highlights the need for rigorous infection control strategies [1]. Standard operating procedures, including hygiene protocols, personal protective equipment, and patient flow control, are critical to limiting infections [1,2]. A key component of these procedures is the early identification and segregation of infectious or potentially infectious patients, often achieved via isolation in dedicated units [1,2]. The use of these units help protect other animals, clients, and hospital personnel, but frequently operate under constraints such as limited staffing, space, or diagnostic capacity [1,3]. As a consequence, rapid and effective triage becomes essential to allocate resources appropriately and prioritize care. In this context, predictive models have gained prominence as tools to support clinical decision-making, particularly for infection risk assessment and management in both human and veterinary settings [4,5].

Feline Immunodeficiency Virus (FIV) and Feline Leukemia Virus (FeLV) are among the most common feline retroviruses worldwide [6,7] and, therefore, pose a major threat to optimal infectious disease management. FIV, a lentivirus, is primarily transmitted through bite wounds and leads to lifelong infection [8,9]. FeLV, a gammaretrovirus, spreads through close contact, especially via saliva and other secretions [10]. Both viruses are associated with nonspecific clinical signs, including anemia, immunodeficiency, comorbidities and coinfections [6,11]. Despite the decline in prevalence due to improved testing and prevention, feline retroviral infections remain a significant concern in veterinary practice, affecting up to 55% of the cats referred to our Biological Isolation and Containment Unit (BICU) in Portugal [7,11,12]. In this unit, it was found that the risk of infection is linked to adult age, male sex (particularly if intact), outdoor access, group housing, and other co-morbidities [13,14]. While confirmatory tests such as Polymerase Chain Reaction (PCR), Western blot for FIV, or Indirect Fluorescent Antibody (IFA) for FeLV are recommended in uncertain cases, ELISA and immunomigration remain the most accessible frontline diagnostics [15]. However, diagnosing FeLV infection is challenging because the virus can establish different outcomes—progressive, regressive, abortive, or focal—that are not always reliably distinguished by point-of-care tests, requiring molecular confirmation in some cases [12].

A complementary approach to traditional testing is to build data-driven triage models that take advantage of epidemiological information on risk factors, history, clinical signs and laboratory results, to help clinicians in the subsequent disease diagnosis. Such a strategy has been attempted to support the diagnosis of non-effusive infectious peritonitis cases, with promising results [16]. Ultimately, the integration of these data-driven triage models into the admitting workflow would help rapidly stratify incoming cats by FIV or FeLV risk using routinely collected data. A high-risk score for infection would trigger immediate confirmatory testing and prompt a response to place patient in BICU for containment and early treatment, whereas a low-risk score would trigger admission to the general ward without overburdening isolation resources.

It is in this context that the present study aimed at assessing the feasibility of building a data-driven triage model to help detecting FIV and FeLV infections before admitting cats to a veterinary hospital. The study is a follow-up of our recent attempt to improve infectious diseases screening in our BICU [3]. Our previous report was based on routinely collected data for detecting canine parvovirosis, leptospirosis, canine distemper and multidrug-resistant bacterial infections, using a case–control study design. Here, we adopted a similar study design to detect these two common infections in domestic cats.

## 2. Materials and Methods

### 2.1. Study Population

The population under study refers to all cats admitted to our BICU between October 2013 and December 2022. In general, BICU at the Faculty of Veterinary Science, University of Lisbon, is a multispecies facility for the hospitalization of animals either with suspected or confirmed infectious diseases. The facility is separated from the teaching hospital (TH). It has two hospitalization wards for dogs and two for cats, each one of them with a capacity for four animals. The facility is equipped with high-efficiency particulate air (HEPA) filters, a video surveillance system, personal protective equipment, standard operating procedures and operates under negative pressure.

Upon arrival at the hospital, all feline patients undergo a routine triage by a veterinary nurse. If an infectious disease is suspected based on clinical signs or history, the patient is referred for consultation at the BICU. During this consultation, the clinical history is reviewed and diagnostic testing, including retroviral testing, is recommended when appropriate. As the costs of these tests are covered by the owners and can represent a significant expense, some owners may decline testing even when clinically advised. In cases where hospitalization is required, cats are admitted directly to the BICU; if further testing subsequently rules out infectious disease, the patient may be transferred to the general hospitalization ward within the teaching hospital (VTH-FMV). This structured triage and referral process is designed to optimize infection control by isolating potentially contagious animals early in the clinical workflow. In this context, a model based on routinely collected clinical data could provide preliminary support for prioritizing animals with higher probability of retroviral infection and may also help clinicians communicate the importance of confirmatory testing to owners.

### 2.2. Participants and Data Collection

We compiled health records from the study period, stored in the TH management software Guruvet^®^ (Veterinary Hospital Management Software, Version 1.1.4, Gurusoft, Lisbon, Portugal) and Qvet^®^ (Veterinary Clinical Management Software, Web Edition Version 25, QSOFT T.I., Porto, Portugal). The records were then introduced in a Microsoft Excel^®^ (Version 16.97.2) spreadsheet by BICU staff for subsequent analysis.

The database accounts for health records of 1211 hospitalized cats in total. Of these, 344 cats had no record of retrovirus testing and were excluded. The remaining 867 cats underwent retrovirus testing: 165 tested FIV-positive and 161 tested FeLV-positive, including 27 co-infected with both viruses. See Section 2.3 for details on the laboratory procedure used for confirmation of these infections. After excluding animals with missing predictor data (complete case scenario), 134 FIV-positive and 126 FeLV-positive cats remained.

The population of FIV and FeLV-positive cats admitted to the BICU included both clinically sick animals and those presenting with milder or nonspecific signs, reflecting the routine case mix of suspected infectious disease patients. For the purpose of this study, cats were classified as FIV or FeLV-positive according to their serological test results obtained at admission or during hospitalization, regardless of whether retroviral infection itself or another concomitant disorder was the primary reason for hospitalization. Control group cats were defined as retrovirus-negative according to diagnostic testing [17,18] but were not necessarily clinically healthy. This approach reflects the clinical reality of hospital admissions at BICU. Moreover, given that FIV-infected cats may remain asymptomatic for extended periods and regressive FeLV infections may not be detected by ELISA or immunomigration tests alone, some degree of misclassification cannot be excluded. A total of 504 retrovirus-negative cats were identified and used as the control group in both the FIV and FeLV analyses.

Overall, the analysis included data from 640 unique cats, representing 52.8% of the total database. Flow diagrams of the FIV and FeLV dataset construction are provided in the Appendix A.

### 2.3. Sample Testing

Different diagnostic tools were used during the study period, depending on clinical context and resource availability. In routine diagnostic workflows, Enzyme-Linked Immunosorbent Assays (ELISA) performed at the Virology and Immunology Laboratory of the Faculty of Veterinary Medicine, University of Lisbon (VIL-FMV-ULisbon) were the standard method for detecting FIV and FeLV infection. These included Viracheck^®^ [FeLV and FIV ELISA Test] (*n* = 606) (Synbiotics Corporation, San Diego, CA, USA), which had a sensitivity of 92.6% for FIV and 94.9% for FeLV, with a specificity of 99.8% for FIV and 98.4% for FeLV [19]. From November 2021 onward, Vetline^®^ [FIV Antibody Test] (*n* = 124) (Biovet Inc., Saint-Hyacinthe, QC, Canada) was introduced for FIV detection. According to the manufacturer’s performance report this test has a diagnostic sensitivity of 95.5% and a specificity of 96.3% [20].

Rapid immunomigration-type assays WITNESS^®^ [FeLV-FIV Test] (*n* = 93) (Zoetis Inc., Parsippany, NJ, USA) were also available and were typically used in consultations for rapid screening, given their ease of use and short turnaround. In cases where these results required confirmation or clarification, ELISA testing was subsequently performed. The sensitivity and specificity of these tests were 93.8% and 93.4% for FIV, and 92.9% and 96.5% for FeLV, respectively [21]. In some cases, additional confirmatory testing with PCR (*n* = 15) or Western blot was performed to confirm or clarify ELISA results. However, these tests were not applied routinely. For this study, when confirmatory testing was applied, the final confirmed result was considered for analysis.

Complete blood counts (CBC) were performed using EDTA-anticoagulated blood samples collected via jugular or cephalic venipuncture. Analyses were conducted using automated hematology analyzers, specifically the IDEXX ProCyte One^®^ (IDEXX Laboratories, Inc., Westbrook, ME, USA) and Zoetis VETSCAN HM5^®^ (Zoetis Inc., USA). Hematocrit values were obtained from automated complete blood counts (CBCs) using calculated hematocrit based on red blood cell indices (RBC × MCV), as per standard analyzer output. Although direct methods such as microhematocrit centrifugation or hemoglobin-based estimation are considered gold standards, they were not consistently documented across clinical records and were therefore excluded from analysis. Calculated hematocrit was the most consistently reported red cell parameter and ensured greater data uniformity, minimized inter-operator variability, and maximized case inclusion—an essential factor in this retrospective study design.

### 2.4. Variables

A set of routinely asked questions during standard hospitalization consultations were used to assess triage-related factors potentially associated with FIV and FeLV. These questions are part of general clinical intake and applicable to various diseases; the variables retained in the analysis were selected based on biological plausibility. Continuous variables were converted into categorical variables to maintain consistency with other variables in the model and to facilitate easier interpretation of results in a clinical context. Recorded data included breed (breed or mixed-breed), sex and neuter status (intact or neutered male and intact or neutered female), number of cats in the household (single cat household or multi-cat household [22], origin (street, shelter, and breeder), lifestyle (outdoor access or strictly indoor), concomitant disorders/diseases (includes any additional diseases infectious or non-infectious that did not require hospitalization in the BICU, such as chronic kidney disease, tick-borne diseases), age groups (young [<2 years], adult [≥2 and <10 years], and geriatric [≥10 years]), according to the Feline Life Stage Guidelines [23]. Vaccination status (updated or not updated), corresponding to the 2020 AAHA/AAFP Feline Vaccination Guidelines [24]. Using CBC of the admission day (approximately 48 h) the total leukocyte count was categorized as low if >5.5 × 10^9^/L, normal 5.5–19.5 × 10^9^/L or high >19.5 × 10^9^/L [25]. Likewise, the hematocrit was categorized as low if <30%, normal 30–45%, and high if >45% [25].

### 2.5. Statistical Analysis

Descriptive analysis were conducted for FIV, FeLV-infected cats, and their controls hospitalized at the BICU using R software (Version 4.2.2) (R Foundation for Statistical Computing, Vienna, Austria) on a MacBook Pro (Apple Inc., Cupertino, CA, USA), macOS Ventura 13.6.5, 16 GB RAM, M1 Pro processor. For simplicity, continuous quantitative variables as described above were converted to categorical variables such as age, leukocyte count, and hematocrit count. Categorical variables were summarized as frequencies and percentages. Cases with missing data were excluded to maintain data integrity and analytical accuracy.

To infer whether FIV infections were significantly associated with FeLV infections, we used the Chi-squared test with Yates’ continuity correction and Fisher’s exact test for 2 × 2 contingent tables (see Appendix A). Given there was no significant association between these infections, we analyzed the data of these two infectious separately.

To analyze data of each infection, we performed Pearson’s chi-square tests and simple logistic regression models to assess differences in key demographic and clinical variables, including age group, sex-neuter status, number of cats in the household, lifestyle (indoor vs. outdoor), leukocyte count, and hematocrit levels. In this analysis, we used a significance level of 5%. We decided to exclude from this analysis all variables included in the database with more than 10% missing data, as done elsewhere [26]; this applied specifically to origin, vaccination status and other CBC parameters.

Variables that showed significant differences between cases and controls (*p* < 0.05) were considered potential confounders and were included in logistic regression models as covariates [27].

Logistic regression models were estimated via the maximum likelihood method. Potential interactions between explanatory variables were assessed, but none were significant at the level of 5%. The subsequent evaluation of model performance was donevia the estimation of the area under the receiver operating characteristic (ROC) curve (AUC) using the pROC package (version 1.18.0) [28]. Optimal sensitivity and specificity thresholds were determined using the ROC01 criterion (i.e., point associated with the minimal distance between the ROC curve and the point of perfect classification) estimated by the optimal.cutpoints package (version 1.1.5) [29]. Additionally, positive and negative predictive values (PPV and NPV, respectively) were calculated at each optimal cutoff. Finally, the models were also assessed by the Hosmer–Lemeshow goodness-of-fit test, as available in the ResourceSelection package (version 0.3.6) [30]. In this test, *p*-values > 0.05 were indicative of evidence for a good fit of the tested model to the data.

### 2.6. Ethics

At the time of each hospital admission, owners were asked to sign a written informed consent form, allowing the collected data for future research purposes. Data was accessed or analyzed after ethical approval by the Ethics Committee of the Faculty of Veterinary Medicine, University of Lisbon (Protocol reference 014/2023).

## 3. Results

### 3.1. Basic Characteristics of the FIV and FeLV-Infected Cats and Their Controls

The initial database consisted of 1211 cats; all entries were independently reviewed and validated by a second investigator to ensure accuracy before analysis, and data entry errors or inconsistencies were checked and corrected during this process. Of these, 165 and 161 cats were confirmed to have FIV and FeLV infections, respectively. We excluded the patients from the analysis if they had missing values in any of the covariates; therefore, the new data frame contained 134 FIV-infected cats and 126 FeLV-infected cats (complete case scenario).

Table 1 displays the basic characteristics of the FIV- and FeLV-infected cats and the respective controls.

FIV-seropositive cats were predominantly adult, mixed-breed, neutered males with outdoor access and concomitant disorders. There were statistically significant associations between FIV infection and age group, sex-neuter status, lifestyle, and the presence of concomitant disorders. Adult (estimate = 1.636, *p* < 0.001) and senior cats (estimate = 2.149, *p* < 0.001), intact males (estimate = 0.757, *p* = 0.031), and cats with outdoor access (estimate = 2.324, *p* < 0.001) seemed to positively contribute to these associations, as did the presence of concomitant disorders (estimate = 0.777, *p* = 0.001) (Table 2). Other variables were not significantly associated with FIV status.

FeLV-seropositive cats were also predominantly mixed-breed, adult animals with concomitant disorders. There was evidence for significant associations between FeLV infection and breed, age group, hematocrit, and concomitant disorders. Mixed-breed cats (estimate = 1.143, *p* = 0.031), cats aged 2–9 years (estimate = 0.616, *p* = 0.016), low hematocrit values (estimate = 0.886, *p* = 0.017), and the presence of concomitant disorders (estimate = 0.497, *p* = 0.036) seemed to contribute to these associations (Table 3). The remaining variables (sex-neuter status, lifestyle, cohabitant cats and leukocyte classification) were not significantly associated with FeLV seropositivity.

### 3.2. Prediction of FIV Status Infections Using Logistic Regression

The final logistic regression model for FIV seropositivity status showed age as a positive and strong predictor (Table 2). In fact, both adult (≥2 and <10 years) and senior cats (≥10 years) had a positive and significant effect in the logistic regression model. Likewise, outdoor access and concomitant disorders were also found to have a positive effect on the likelihood of FIV seropositivity (*p* < 0.05). The remaining putative triage factors (mixed-breed cats, living with multiple cats, neutered males, hematocrit and leukocyte classification) were not statistically significant in our dataset, but they were retained in the final model because their exclusion reduced predictive performance. Furthermore, these variables are supported in the literature as clinically relevant factors, which strengthens the interpretability of the model.

In terms of performance, the estimated AUC of the model was 0.70 (95% CI = [0.65–0.75]) (Figure 1). Using the optimal cutpoint based on the Youden Index, the sensitivity and specificity estimates were 63% and 69%, respectively, which suggested that this model had similar performance on predicting cases and non-case and illustrate the balance between minimizing missed infections and avoiding unnecessary admissions, a key consideration in triage.

When the ROC01 criterion was used alternative to calculate the optimal cutoff, the above estimates for the sensitivity and specificity almost did not vary (65.7% and 63.5%, respectively). The model also achieved a PPV of 32.4% and a NPV of 87.4%, supporting its utility for ruling out infection among predicted negatives (see Appendix A).

Finally, the model has a good agreement with the data according to the Hosmer–Lemeshow goodness-of-fit test (*p* = 0.878).

### 3.3. Prediction of FeLV Infections Using Logistic Regression

The final logistic regression models suggested that the most important factors for triaging FeLV seropositivity were breed, concomitant disorders and a low hematocrit count. Some variables were not statistically significant in our dataset, but they were retained in the final model because their exclusion reduced predictive performance. In addition, these variables are consistently reported in the literature as relevant factors for FeLV seropositivity, supporting their inclusion to preserve the clinical interpretability of the model.

The final logistic regression model showed an AUC of 0.69 (95% CI = [0.64–0.74]), an estimate similar to the one obtained for predicting FIV infection status (Figure 1). Using the ROC01 criterion for calculating optimal sensitivity and specificity, the corresponding estimates were 65.1% and 60.5%, respectively. The associated PPV and NPV were 29.2% and 87.4%, respectively, suggesting that the model was more effective at ruling out FeLV infection when negative (see Appendix A). The Hosmer–Lemeshow test also provided evidence for a good fit of the model to the data (*p* = 0.969).

## 4. Discussion

### 4.1. FIV Infections

In our analysis of FIV seropositivity, outdoor access, presence of concomitant diseases and older age groups (≥2 years) were significantly associated with increased likelihood of FIV seropositivity, while no statistically significant associations were observed for sex-neuter status, breed, hematocrit, or leukocyte count.

Regarding the variable breed, there was no significant association. A similar lack of breed effect was reported in a large-scale North American survey [14]. Other researchers also found no evidence of breed predisposition to FIV seropositivity [11,30]. However, only five FIV-positive cats in our dataset were purebred. This very limited number reduces the statistical power to detect breed-specific associations. As such, conclusions regarding breed and FIV infection must be interpreted with caution.

Previous studies have consistently reported an association between intact male cats and FIV seropositivity, attributing the risk to increased territorial aggression and bite-related transmission [8,11,15,30,31,32,33]. However, in our revised analysis, no statistically significant association was found between sex-neuter status and FIV seropositivity. This discrepancy may reflect sample variability or the influence of other confounding variables such as lifestyle or age group. Nevertheless, the biological plausibility of this association remains supported by the literature, particularly the higher risk behaviors observed in intact males [8,15].

Consistent with other research studies [15,30], our analysis also observed that cohabitation with other cats did not significantly influence admission to the BICU with FIV. This is particularly evident in a study conducted by Litster, in which no transmission of FIV was recorded over several years among 130 uninfected cats living with eight FIV-infected cats [34].

A correlation was found between outdoor access and FIV seropositivity. Outdoor access is a possible triage-related factor, because free-roaming cats are more likely engage in territorial disputes and be exposed to infected conspecifics [11]. Levy et al. identified outdoor lifestyle as one of the strongest risk factors for FIV positivity, with an adjusted odds ratio of approximately 3.0 [14].

FIV is associated with progressive immunosuppression; therefore, FIV-infected cats may be predisposed to secondary and opportunistic infections [8,15]. This observation is supported by Levy et al., who found higher FIV seroprevalence among cats presenting with systemic illness [14]. Nevertheless, clinical immunodeficiency may take years to manifest. Not all infected cats are immunosuppressed, especially in the early stages of infection. In our study, FIV seropositivity was significantly associated with concomitant diseases. This association may reflect the clinical impact of FIV when secondary conditions arise [18], although in some cases the concomitant disease itself may have been the primary reason for hospitalization.

While bone marrow suppression can occur in FIV infections, it is noteworthy that this phenomenon is more commonly associated with FeLV-infected cats [8,12,15,35]. In FeLV cases, bone marrow disorders, including anemia, are often observed [12]. However, in FIV infections, the acute phase is characterized by mild neutropenia, and in later stages, lymphopenia may occur [36,37]. Despite these hematological changes, our study did not find a significant association between leukocyte or hematocrit levels and FIV seropositivity. This result is consistent with the general understanding that FIV is typically less pathogenic than FeLV [12,38]. Other hematological values were excluded as they had more than 10% missing data. We applied a 10% cutoff for simplicity, opting for exclusion rather than imputation. FIV-infected cats can remain asymptomatic or mildly symptomatic for an extended period, and the progression of the disease varies among individuals [15].

FIV seropositivity was statistically relevant in cats between the ages of ≥2 and <10 years and ≥10 years, respectively, compared to cats with less than 2 years. This result was consistent with previous observations that an increase in age leads to susceptibility of contracting FIV [14,15,30]. This increase in age can be explained by the fact that FIV-positive cats can remain asymptomatic for many years and only express clinical signs later in life when diagnosed at that stage [30].

### 4.2. FeLV Infections

The multivariable analysis of FeLV seropositivity revealed significant associations with mixed-breed status, the presence of concomitant diseases, and reduced hematocrit values. By contrast, sex-neuter status, outdoor access, multi-cat households, and leukocyte counts were not independently associated. The effect of age is significant in the univariate analysis, but this did not persist in the final model.

Typically, FeLV does not have a breed predisposition; however, in this study, an association between mixed-breed cats and FeLV seropositivity was found. According to Hofmann-Lehmann and Hartmann [10], FeLV is commonly found in mixed-breed cats because purebreds tend to be kept indoors, and there is greater awareness from the owners to test their animals. This aligns with the broader understanding that indoor management and proactive healthcare in purebred cats may have a lower risk of infection. Similarly, Studer et al. found that pedigree status was strongly protective, with mixed-breed cats showing markedly higher odds of FeLV positivity across Europe [13]. Nevertheless, only four FeLV-seropositive cats were purebred, which limits our ability to assess the role of breed in FeLV seropositivity. The apparent association with mixed-breed status may partially reflect population demographics rather than true biological predisposition. Yet, other studies [11,30] have not found any association.

Concerning sex and neuter status previous researchers indicated that FeLV infection rates were roughly equal between sexes, because it can spread among queens and their kittens and among cats living in close quarters, making transmission possible regardless of sex [12,31]. Others have suggested that male cats are at higher risk [11,12,30,31,39,40]. The impact of neutering remains unclear and appears to vary across different cat populations, such as owned, stray, or shelter-housed animals. In our analysis, no significant association was found between sex-neuter status and FeLV seropositivity.

Although FeLV is considered a “social” disease [12,31], living in a multi-cat household was not significantly associated with FeLV-seropositivity. According to Gleich et al. and Studer et al. [13,31], heightened disease awareness has led to more proactive testing and cautious integration of new cats in a multi-cat household.

The non-significant association between outdoor access and FeLV seropositivity challenges the traditional view of outdoor exposure as a risk factor. While previous literature suggests that outdoor access increases the risk of infection [12], other contemporary studies had the same result as ours [11,30,31]. Levy et al. [14] suggests that indoor cats especially in multi-cat households or among adopted cats with unknown histories.

Similarly to FIV, FeLV suppresses the immune system; however, FeLV-related immune dysfunction may begin earlier in the course of infection. While leukocyte counts may remain within normal ranges, immune function is often impaired, increasing susceptibility to secondary infections [11,12,35]. Our study found a significant association between concomitant diseases and FeLV seropositivity, which is supported by Levy et al. [14], who reported elevated FeLV seroprevalence in clinically sick cats, especially those with signs such as anemia, and anorexia. However, this association should be interpreted with caution, as in some cases the concomitant disease itself may have been the primary reason for hospitalization rather than FeLV infection per se.

While FeLV is commonly associated with bone marrow suppression, the most prevalent hematological abnormality observed in our study, as well as in the literature is anemia [12,41]. Our findings determined that a lower hematocrit was significantly associated with FeLV seropositivity. It is important to note that while bone marrow suppression can affect other blood cell lines such as leukocytes, our study did not find a significant association between leukocyte classification and FeLV-seropositive cats. It is important to consider that calculated hematocrit values may be affected by pathological alterations in erythrocyte morphology commonly observed in retroviral infections. In FeLV-seropositive cats, changes such as anisocytosis, poikilocytosis, and macrocytosis are frequently reported and can influence MCV, potentially leading to deviations in calculated hematocrit that may not fully reflect the true extent of anemia. Moreover, concomitant disorders could also alter CBC values, meaning that these associations should be interpreted with caution. These virus-associated distortions are a known limitation of analyzer-derived indices and may partly explain variability observed in the hematological profiles. Although this parameter was selected based on data consistency, the potential analytical impact of erythrocyte abnormalities in infected cats should be kept in mind when interpreting associations. This issue further supports the need for prospective studies employing direct measurements of anemia when feasible [42].

In the univariate analysis, adult cats (aged ≥2 years) showed a statistically significant association with FeLV seropositivity. However, this association was not retained in the final multivariable model, suggesting that age alone may not independently predict FeLV seropositivity when other covariates are considered. This partial association aligns with previous findings indicating an age-dependent susceptibility, where younger cats are more prone to progressive FeLV infection due to immature immune responses [12]. Nonetheless, recent studies, indicate that adult cats are more frequently diagnosed with FeLV [12,14]. This trend may be attributed to heightened awareness, leading owners to test their cats more regularly and seek early medical intervention, thus potentially extending the cats’ lifespans [12]. In our study, similar age distributions between infected and control cats may have contributed to the lack of an independent association in multivariable analysis.

Additionally, FeLV infection presents in different clinical forms, including progressive, regressive, abortive, and focal infections [12]. In particular, regressive FeLV infection is characterized by an initial transient viremia that is eventually controlled by the host’s immune response, leading to undetectable or low levels of circulating antigen. These cats may test negative with point-of-care (POC) antigen tests such as ELISA, despite harboring proviral DNA in their tissues [12]. Regressive infections can remain latent for extended periods and may only become detectable during routine testing later in life or under conditions of immunosuppression [43]. This phenomenon may partly explain the detection of FeLV in older cats within our dataset and should be considered when interpreting infection timing or progression. However, since our study relied solely on POC and ELISA tests and did not employ molecular confirmation (e.g., PCR), we could not distinguish between progressive and regressive infections. Future studies should explore this distinction using confirmatory diagnostics to better understand the dynamics of FeLV infection stages.

### 4.3. Limitations of the Study

As a retrospective proof-of-principle study, the present models are not intended for immediate clinical application. The retrospective nature of this study also introduces limitations, most importantly the problem of missing data and the inability to consider other triage variables beyond those routinely collected in the BICU. Nonetheless, these limitations represent an opportunity to reflect on how clinical practice and research can be better integrated.

A possible limitation is the misclassification of regressively FeLV-seropositive cats as FeLV-negative controls, due to the lack of confirmatory PCR testing. Regressive infections, which are antigen-negative but provirus-positive, cannot be detected by antigen-based assays such as ELISA or immunomigration tests [43]. This may have introduced classification bias, potentially underestimating true positives and compromising model specificity. However, the diagnostic methodology adopted reflects real-world screening practices in clinical hospital settings, where PCR testing is not routinely used in initial triage workflows.

Additionally, heterogeneity in diagnostic testing might introduce variations in the sensitivity and specificity of the respective diagnosis over time. In fact, multiple ELISA tests and rapid immunomigration kits were used during the study period, each test with slightly different performance characteristics. Although the documentation of all tests suggested sensitivities and specificities above 90%, variation in test methodology and diagnostic accuracy may have influenced classification, especially in borderline cases. It should also be noted that concomitant diseases may also have influenced hematological values, representing an additional study limitation.

The study design also restricted the selection of controls and cases to animals with definitive diagnostic results, which were influenced by the clinical status of the animals and the economic considerations of the owners.

Finally, the absence of external validation limits the application of the models beyond the realm of BICU and its data. In this perspective, similar studies should be conducted with data from other units in order to understand whether the estimated triage models used here are in fact general, therefore, usable in units with similar clinical scope. In this line of thought, we continue collecting data in BICU in order to validate the performance of the estimated models further.

## 5. Conclusions

This study constructed triage models for FIV and FeLV infections using simple logistic regression models based on routinely collected clinical data. The construction of these models is the first step towards a more data-driven triage approach. In our BICU, key triage factors were adult and senior cats, outdoor access for FIV seropositivity, and anemia for FeLV, highlighting the importance of restricted outdoor access, and proactive monitoring for secondary conditions. These models should therefore be viewed as preliminary, no-cost decision-support tools that may assist clinicians in prioritizing cases and encouraging owner compliance with confirmatory diagnostic testing. They are not substitutes for point-of-care assays or ELISAs, but they provide a theoretical foundation for developing more robust clinical scoring protocols, particularly in settings with limited resources.

Follow-up work should be conducted towards in order to perform model validation with data external to the unit, to improve the models and their sensitivity/specificity, and to develop a computer application based on the constructed models for the respective deployment into the unit. Although the variables used are part of standard feline history and examination, the case-mix in our study was shaped by the specific referral profile of the BICU, with a higher prevalence of infectious disease suspicions. This context may influence the relative predictive weight of individual factors. Therefore, we recommend that similar modeling exercises be undertaken in other units, allowing validation and, if necessary, the adaptation of the models to the characteristics of their own patient populations.

## Figures and Tables

**Figure 1 vetsci-12-00902-f001:**
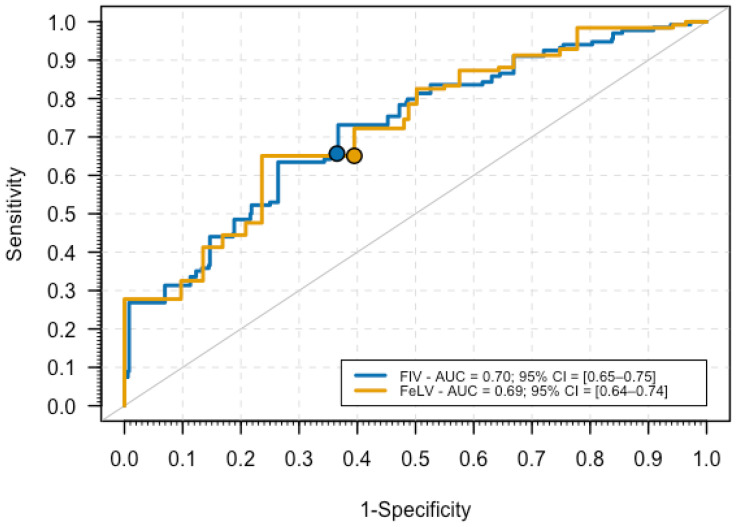
ROC curves for the final regression models of FIV and FeLV infections, where the dots represent the optimal sensitivity/specificity according to the ROC01 criterion.

**Table 1 vetsci-12-00902-t001:** Basic characteristics of FIV- and FeLV-seropositive cats and their controls.

Characteristics	Categories	FIV	Controls	FeLV
		Case, *n* = 134 Cats (%)	*n* = 504 Cats (%)	Case, *n* = 126 Cats (%)
Breed	Purebred	5 (3.7%)	47 (9.3%)	4 (3.2%)
	Mixed-breed	129 (96.3%)	457 (90.7%)	122 (96.8%)
Sex and Neuter Status	Neutered Male	71 (53.0%)	199 (39.5%)	39 (31.0%)
	Intact Male	20 (14.9%)	72 (14.3%)	23 (18.3%)
	Neutered Female	29 (21.6%)	152 (30.2%)	39 (31.0%)
	Intact Female	14 (10.4%)	81 (16.1%)	25 (19.8%)
Number of cats in the household	Single Cat	58 (43.3%)	210 (41.7%)	57 (45.2%)
	≥2 Cats	76 (56.7%)	294 (58.3%)	69 (54.8%)
Lifestyle	Indoor	52 (38.8%)	314 (62.3%)	72 (57.1%)
	Outdoor	82 (61.2%)	190 (37.7%)	54 (42.9%)
Concomitant Disorders/Diseases	No	23 (17.2%)	156 (31.0%)	27 (21.4%)
	Yes	111 (82.8%)	348 (69.0%)	99 (78.6%)
Age Groups	<2	13 (9.7%)	132 (26.2%)	25 (19.8%)
(Years)	≥2 and <10	82 (61.2%)	231 (45.8%)	81 (64.3%)
	≥10	39 (29.1%)	141 (28.0%)	20 (15.9%)
Leukocytes Classification	High ^1^	28 (20.9%)	120 (23.8%)	25 (19.8%)
	Normal ^2^	79 (59.0%)	285 (56.5%)	69 (54.8%)
	Low ^3^	27 (20.1%)	99 (19.6%)	32 (25.4%)
Hematocrit Classification	High ^4^	16 (11.9%)	65 (12.9%)	12 (9.5%)
	Normal ^5^	77 (57.5%)	276 (54.8%)	45 (35.7%)
	Low ^6^	41 (30.6%)	163 (32.3%)	69 (54.8%)

Reference ranges: ^1^ high > 19.5 × 10^9^/L; ^2^ normal: 5.5–19.5 × 10^9^/L; ^3^ low < 5.5 × 10^9^/L; ^4^ high > 45%; ^5^ normal: 30–45%; ^6^ low < 30%.

**Table 2 vetsci-12-00902-t002:** Simple and multiple regression models for predicting the probability FIV seropositivity (*n* = 638).

Characteristics	Categories	Simple Logistic Regression			Multiple Logistic Regression		
		Estimate	Std Error	*p*-Value	Estimate	Std Error	*p*-Value
Breed	Purebred (reference)	—	—	NA	—	—	NA
	Mixed-breed	0.976	0.481	0.042	0.627	0.497	0.207
Sex and Neuter Status	Intact Female (reference)	—	—	NA	—	—	NA
	Intact Male	0.474	0.384	0.217	0.606	0.413	0.142
	Neutered Female	0.098	0.353	0.779	−0.306	0.372	0.411
	Neutered Male	0.724	0.321	0.024	0.272	0.341	0.425
Number of cats in the household	1 (reference)	—	—	NA	—	—	NA
	>1	−0.066	0.196	0.736	—	—	NA
Lifestyle	Indoor (reference)	—	—	NA	—	—	NA
	Outdoor	0.958	0.2	<0.001	0.864	0.208	<0.001
Concomitant Disorders/Diseases	No (reference)	—	—	NA	—	—	NA
	Yes	0.772	0.246	0.002	0.674	0.269	0.012
Age Groups (Years)	<2 (reference)	—	—	NA	—	—	NA
	≥2 and <10	1.282	0.318	<0.001	1.185	0.351	<0.001
	≥10	1.033	0.342	0.003	0.974	0.386	0.011
Classification of Leukocyte	High ^1^ (reference)	—	—	NA	—	—	NA
	Normal ^2^	0.172	0.245	0.483	—	—	NA
	Low ^3^	0.156	0.302	0.605	—	—	NA
Classification of Hematocrit	High ^4^ (reference)	—	—	NA	—	—	NA
	Normal ^5^	0.125	0.307	0.684	—	—	NA
	Low ^6^	0.022	0.329	0.948	—	—	NA

^1^ high > 19.5 × 10^9^/L; ^2^ normal: 5.5–19.5 × 10^9^/L; ^3^ low < 5.5 × 10^9^/L; ^4^ high > 45%; ^5^ normal: 30–45%; ^6^ low < 30%.

**Table 3 vetsci-12-00902-t003:** Simple and Multiple regression models for predicting the probability of FeLV seropositivity (*n* = 630).

Characteristic	Categories	Simple Logistic Regression			Multiple Logistic Regression		
		Estimate	Std Error	*p*-Value	Estimate	Std Error	*p*-Value
Breed	Purebred (reference)	—	—	NA	—	—	NA
	Mixed-breed	1.143	0.531	0.031	1.221	0.541	0.024
Sex and Neuter Status	Intact Female (reference)	—	—	NA	—	—	NA
	Intact Male	0.034	0.331	0.917	—	—	NA
	Neutered Female	−0.185	0.291	0.525	—	—	NA
	Neutered Male	−0.454	0.288	0.115	—	—	NA
Number of cats in the household	1 (reference)	—	—	NA	—	—	NA
	>1	−0.145	0.201	0.468	—	—	NA
Lifestyle	Indoor (reference)	—	—	—	—	—	NA
	Outdoor	0.215	0.202	0.288	—	—	NA
Concomitant Disorders/Diseases	No (reference)	—	—	NA	—	—	NA
	Yes	0.497	0.238	0.036	0.522	0.254	0.04
Age Groups (Years)	<2 (reference)	—	—	NA	—	—	NA
	≥2 and <10	0.616	0.254	0.016	0.408	0.266	0.125
	≥10	−0.289	0.324	0.372	−0.617	0.344	0.073
Classification of Leukocyte	High ^1^ (reference)	—	—	NA	—	—	NA
	Normal ^2^	0.15	0.258	0.56	—	—	NA
	Low ^3^	0.439	0.299	0.142	—	—	NA
Classification of Hematocrit	High ^4^ (reference)	—	—	NA	—	—	NA
	Normal ^5^	−0.124	0.353	0.725	−0.050	0.358	0.888
	Low ^6^	0.886	0.372	0.017	0.897	0.352	0.011

^1^ high > 19.5 × 10^9^/L; ^2^ normal: 5.5–19.5 × 10^9^/L; ^3^ low < 5.5 × 10^9^/L; ^4^ high > 45%; ^5^ normal: 30–45%; ^6^ low < 30%.

## Data Availability

The datasets used and/or analyzed during the current study are available from the corresponding author upon reasonable request.

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
