# Peer review of "Statistical Triage Model for Feline Infectious Diseases in a Veterinary Isolation Unit: The Case of Feline Immunodeficiency and Leukemia Viruses"

_vetsci, 2025, doi:10.3390/vetsci12090902_

Round 1
Reviewer 1 Report
Comments and Suggestions for Authors
In this manuscript, the authors present an early form of a model for determining the risk of FeLV and FIV in their hospital population. This was performed via retrospective data collection, so the authors are at the mercy of what is in the records. I really like the idea of building such a model. However, I found the methods section really confusing and I didn’t understand if the cats were classified according to disease or serology results. Most of the seropositive cats for both viruses were had another disorder/disease, which makes sense if the animals presented to a teaching hospital that doesn’t do much routine veterinary care. Also, I am not sure if the model was built with some cats and then tested or if this is just a description of this cohort of cats. It would be interesting to see how the model performs in the real world. However, that creates bias in the study population. The authors were careful to explain the limitations of some of their results in the discussion section, which is appreciated. The tables were nicely presented and easy to understand.
Title
The paper isn’t really about the feasibility of building a model. The discussion is about the findings of the model you built.
Introduction
Line 80: Is this Portuguese BICU different from the one that you used for data collection?
Generally, it might be useful to talk a little more about the diseases here, rather than in the discussion.
Materials and Methods
Lines 125-127: Please present the results of the internal validation in the results section.
Lines 128-134: This is really my biggest question. Were these cats presented for clinical signs related to the viruses? Or were cats presented for something else, like an abscess, that might reasonably trigger a test for FIV? Another disease process may affect cell counts, etc.
Lines: 159-162: For methods, is it correct that you used the Witness and Vetline tests for screening all cats and then ELISA for some cats? If ELISA was not used as the definition of seropositivity/seronegativity then I don’t see that you have to talk about it in this paper. I’d be interested in any discrepancies (really outside the scope of your article, but I am wondering how those cats were categorized).
Lines 185-187: Is this comparison non-infectious vs other infectious? Or concomitant diseases vs no concomitant disease? It goes back to why the cats were seen at the hospital, ie for clinical signs related to the virus or for chronic renal disease? If the latter, could those diseases have affected the CBC results?
Lines 208-211: Why were origin and vaccine status not included in your analysis?
Lines 216-219: I don’t see a need to justify why you did not perform a certain analysis. Logistic regression is appropriate for your data. Were interactions between the variables assessed?
Results
Line 266, 286: Can you explain including the non-significant variables improves accuracy?
Line 268-270: Sensitivity and specificity are determined using a gold standard test, which would include confirmatory testing. I’m not sure that Se and Sp are relevant here, since the goal was to screen for seropositivity before admitting cats.
Discussion
Lines 301-302: I appreciate that you included this limitation of the breed data.
Lines 327-328: Can you tell that the increased hospitalization rate was due to FIV positivity? For instance, a cat with CRF might be hospitalized for fluids, not for anything related to being FIV positive.
Line 340: I’m sorry to harp on this, but saying “being hospitalized with FIV” implies (to me, anyway) that the cats were hospitalized for FIV and not with some other disease while being seropositive.
Line 438 – 446: There are some typos in this paragraph.
Line 468: I don’t understand why the model needs to be developed at each individual BICU. The variables you collected would be part of a standard cat history. What would be different in the populations that would make this non-generalizable.
Abbreviations
I didn’t see HEV-FMB, which I’m guessing refers to a specific BICU?
References
Minor, but ref 14 is missing the journal name.
Author Response
General Comment: In this manuscript, the authors present an early form of a model for determining the risk of FeLV and FIV in their hospital population. This was performed via retrospective data collection, so the authors are at the mercy of what is in the records. I really like the idea of building such a model. However, I found the methods section really confusing and I didn’t understand if the cats were classified according to disease or serology results. Most of the seropositive cats for both viruses were had another disorder/disease, which makes sense if the animals presented to a teaching hospital that doesn’t do much routine veterinary care. Also, I am not sure if the model was built with some cats and then tested or if this is just a description of this cohort of cats. It would be interesting to see how the model performs in the real world. However, that creates bias in the study population. The authors were careful to explain the limitations of some of their results in the discussion section, which is appreciated. The tables were nicely presented and easy to understand.
Reply: We thank the reviewer for the constructive feedback, comments and suggestions of clarification. Cats were indeed classified according to their serological status (FIV- or FeLV-positive/negative) at admission or during hospitalization, regardless of whether retroviral infection itself or another concomitant disorder was the primary reason for hospitalization. This point was clarified in the “Participants and Data Collection” section (pages 3–4, lines 134–144) and discussed in the “Discussion and Conclusions”. We also acknowledge the limitations of working with retrospective data, including possible misclassification and the influence of concomitant diseases on hematological values. These limitations were now expanded in the Limitations of the study (Page 14, lines 460–465 and 479-481). Finally, we reorganized the Methods section to provide a better description of the diagnostic workflow (ELISA, immunomigration tests, and confirmatory testing).
Specific comments
Comments 1: Title - The paper isn’t really about the feasibility of building a model. The discussion is about the findings of the model you built.
Reply: We thank the reviewer for the comment. Similar comment was raised by another reviewer. As such, we have modified the title as suggested by this reviewer: “Statistical triage model for feline infectious diseases in a veterinary isolation unit: The case of feline immunodeficiency and leukemia viruses.”
Comments 2: Introduction - Line 80: Is this Portuguese BICU different from the one that you used for data collection? Generally, it might be useful to talk a little more about the diseases here, rather than in the discussion.
Reply: We thank the reviewer for this comment. To clarify, the BICU described in the Introduction is the same facility from which the data were collected. We have revised the text to make this clearer (Page 2, lines 74–77) says: “Despite the decline in prevalence due to improved testing and prevention, feline retroviral infections remain a significant concern in veterinary practice, affecting up to 55% of the cats referred to our Biological Isolation and Containment Unit (BICU) in Portugal [7,11,12].”Additionally, we have expanded the Introduction to provide more information about FeLV and the complexity of its diagnosis, as also suggested by another reviewer. The revised text (Page 2, lines 82–85) now reads: "However, diagnosing FeLV infection is challenging because the virus can establish different outcomes—progressive, regressive, abortive, or focal—that are not always reliably distinguished by point-of-care tests, requiring molecular confirmation in some cases [12].” A more detailed interpretation of how this diagnostic complexity may have affected our dataset was left to the Discussion section, where potential misclassification and its impact on the results are addressed.
Comments 3: Lines 125-127: Please present the results of the internal validation in the results section.
Reply: We thank the reviewer for this suggestion. We then revised the Methods as follows (Page 4, lines 125–126) now reads: “The records were then introduced in a Microsoft Excel® (Version 16.97.2) spreadsheet by BICU staff for subsequent analysis.” The Results section (Page 6, lines 245-247) now begins as: “The initial database consisted of 1211 cats; all entries were independently reviewed and validated by a second investigator to ensure accuracy before analysis, and data entry errors or inconsistencies were checked and corrected during this process.”
Comments 4: Lines 128-134: This is really my biggest question. Were these cats presented for clinical signs related to the viruses? Or were cats presented for something else, like an abscess, that might reasonably trigger a test for FIV? Another disease process may affect cell counts, etc.
Reply: We thank the reviewer for raising this question. Cats were referred to BICU due to a suspicion or confirmation of infectious disease. Hence, some cats showed clinical signs potentially related to retroviral infection while others presented nonspecific or different clinical conditions that nevertheless warranted retrovirus testing. For the purpose of this study, cats were classified as FIV- or FeLV-positive according to their serological test results obtained during admission or hospitalization, regardless of whether retroviral infection itself or another concomitant disorder was the primary reason for hospitalization. Therefore, the revised text (Pages 4 and 5, lines 134–144) reads: “The population of FIV and FeLV-positive cats admitted to the BICU included both clinically sick animals and those presenting with milder or nonspecific signs, reflecting the routine case mix of suspected infectious disease patients. For the purpose of this study, cats were classified as FIV or FeLV-positive according to their serological test results obtained at admission or during hospitalization, regardless of whether retroviral infection itself or another concomitant disorder was the primary reason for hospitalization. Control group cats were defined as retrovirus-negative according to diagnostic testing [17,18] but were not necessarily clinically healthy. This approach reflects the clinical reality of hospital admissions at BICU. Moreover, given that FIV-infected cats may remain asymptomatic for extended periods and regressive FeLV infections may not be detected by ELISA or immunomigration tests alone, some degree of misclassification cannot be excluded”
In some cases, concomitant diseases may have been the main reason for hospitalization, and such processes could indeed have influenced hematological values. We highlight this aspect of our study in the Discussion (Page 14, lines 481–483), which now reads: “It should also be noted that concomitant diseases may also have influenced hematological values, representing an additional study limitation.”
Comments 5: Lines: 159-162: For methods, is it correct that you used the Witness and Vetline tests for screening all cats and then ELISA for some cats? If ELISA was not used as the definition of seropositivity/seronegativity then I don’t see that you have to talk about it in this paper. I’d be interested in any discrepancies (really outside the scope of your article, but I am wondering how those cats were categorized).
Reply: We thank the reviewer for raising this point, which was also raised by another reviewer. To clarify, ELISA was the standard diagnostic method used in our study. Viracheck® ELISA for both FIV and FeLV was applied during most of the study period and was replaced in 2021 by Vetline® ELISA only for FIV. Rapid immunomigration tests (Witness®) were also used during consultations for a quick screening, and ELISA was subsequently applied in cases requiring confirmation or clarification. In rare cases where ELISA results were inconclusive, additional testing with PCR or Western blot was performed. For this study, the final confirmed result was always used for classification and analysis. We have revised the “Sample Testing” section to clarify of our screening strategy better (Page 4, lines 152–153; 163–165; 167–170):
“In routine diagnostic workflows, Enzyme-Linked Immunosorbent Assays (ELISA)…”;
“… were also available and were typically used in consultations for rapid screening, given their ease of use and short turnaround. In cases where these results required confirmation or clarification, ELISA testing was subsequently performed.”;
“In some cases, additional confirmatory testing with PCR (n=15) or Western blot was performed to confirm or clarify ELISA results. However, these tests were not applied routinely. For this study, when confirmatory testing was applied, the final confirmed result was considered for analysis.”
Comments 6: Lines 185-187: Is this comparison non-infectious vs other infectious? Or concomitant diseases vs no concomitant disease? It goes back to why the cats were seen at the hospital, ie for clinical signs related to the virus or for chronic renal disease? If the latter, could those diseases have affected the CBC results?
Reply: We thank the reviewer for these questions. The variable “concomitant disorders” included any additional diseases (infectious or non-infectious) that did not require hospitalization in the BICU. Cats were admitted to the BICU due to suspicion or confirmation of an infectious disease, but they could also present other co-morbidities, such as chronic kidney disease, tick-borne diseases, or other conditions that did not require isolation. We have now revised the text in the “Variables” subsection (Page 5, lines 193-195): “concomitant disorders/diseases (includes any additional diseases infectious or non-infectious that did not require hospitalization in the BICU, such as chronic kidney disease, tick-borne diseases) …”
We also acknowledge that these concomitant diseases may have influenced CBC results, which is highlighted as a limitation in the Discussion, as already stated in comment 4.
Comments 7: Lines 208-211: Why were origin and vaccine status not included in your analysis?
Reply: We thank the reviewer for this question. As noted in the Methods, variables with more than 10% missing data were excluded. We did not apply imputation methods, as they were beyond the scope of this article, and instead opted for a simpler approach. We have now clarified in the text that this applied specifically to origin, vaccination status, and other CBC parameters. The revised sentence in the Methods section (Page 5, lines 220–222) now reads:
“We decided to exclude from this analysis all variables included in the database with more than 10% missing data, as done elsewhere [26]; this applied specifically to origin, vaccination status and other CBC parameters.”
Comments 8: Lines 216-219: I don’t see a need to justify why you did not perform a certain analysis. Logistic regression is appropriate for your data. Were interactions between the variables assessed?
Reply: We thank the reviewer for this comment. We then removed the justification for not applying alternative models. We also clarified in the Methods that interactions between explanatory variables were assessed during model evaluation. However, no significant interaction terms were identified, and therefore none were retained in the final models. The revised text (Page 5, lines 226–228) now reads: “Potential interactions between explanatory variables were assessed, but none were significant at the level of 5%.”
Comments 9: Line 266, 286: Can you explain including the non-significant variables improves accuracy?
Reply: We thank the reviewer for this comment. The inclusion of non-significant variables in the final models was justified because removing them led to a reduction in overall model performance (lower sensitivity/specificity and area under the ROC curve). In addition, these variables have biological plausibility and are cited in the literature as relevant risk factors, which supports their retention for clinical interpretability. We have then revised the text as follows (Pages 8, Results, lines 276–281):
“The remaining putative triage factors (mixed-breed cats, living with multiple cats, neutered males, hematocrit and leukocyte classification) were not statistically significant in our dataset, but they were retained in the final model because their exclusion reduced predictive performance. Furthermore, these variables are supported in the literature as clinically relevant factors, which strengthens the interpretability of the model.”
It also reads (Page 10, lines 302–306) “Some variables were not statistically significant in our dataset, but they were retained in the final model because their exclusion reduced predictive performance. In addition, these variables are consistently reported in the literature as relevant factors for FeLV infection, supporting their inclusion to preserve the clinical interpretability of the model.”
Comments 10: Line 268-270: Sensitivity and specificity are determined using a gold standard test, which would include confirmatory testing. I’m not sure that Se and Sp are relevant here, since the goal was to screen for seropositivity before admitting cats.
Reply: We thank the reviewer for this thoughtful comment and apologize if our wording was unclear. We agree that sensitivity and specificity are traditionally defined in relation to a gold standard test. In our case, they were reported as standard measures of predictive performance for logistic regression models, complementing the AUC. We believe that presenting sensitivity and specificity remains valuable for characterizing model behaviour at the chosen cutoff. Since the primary goal of these models is to support triage decisions, such as facilitating admission either to the BICU or the general ward and prioritizing confirmatory testing. These measures provide clinically meaningful information on how well the model distinguishes between likely cases and non-cases. and helping to illustrate the model’s potential utility in a triage context. To clarify our point of view, we have now revised the text (Page 8, lines 283–287):
“Using the optimal cutpoint based on the Youden Index, the sensitivity and specificity estimates were 63% and 69%, respectively, which suggested that this model had similar performance on predicting cases and non-case and illustrate the balance between minimizing missed infections and avoiding unnecessary admissions, a key consideration in triage.”
Comments 11: Lines 301-302: I appreciate that you included this limitation of the breed data.
Reply: We thank the reviewer for this positive feedback.
Comments 12: Lines 327-328: Can you tell that the increased hospitalization rate was due to FIV positivity? For instance, a cat with CRF might be hospitalized for fluids, not for anything related to being FIV positive.
Reply: We thank the reviewer for this point. Our data cannot indeed establish causality for hospitalization, as cats may have been admitted due to concomitant conditions unrelated to FIV or FeLV themselves. We have revised the text in both the FIV and FeLV discussion sections to clarify that the observed associations with concomitant disorders reflect correlation rather than direct causation, and that in some cases the comorbidity may have been the primary driver of hospitalization.
The revised text (Page 12, lines 351–354) now reads:
“In our study, FIV seropositivity was significantly associated with concomitant diseases. This association may reflect the clinical impact of FIV when secondary conditions arise [18], although in some cases the concomitant disease itself may have been the primary reason for hospitalization.”
It also now reads (Page 13, lines 409–414):
“Our study found a significant association between concomitant diseases and FeLV seropositivity, which is supported by Levy et al. [14], who reported elevated FeLV seroprevalence in clinically sick cats, especially those with signs such as anemia, and anorexia. However, this association should be interpreted with caution, as in some cases the concomitant disease itself may have been the primary reason for hospitalization rather than FeLV infection per se.”
Comments 13: Line 340: I’m sorry to harp on this, but saying “being hospitalized with FIV” implies (to me, anyway) that the cats were hospitalized for FIV and not with some other disease while being seropositive.
Reply: We thank the reviewer for spotting this oversight from our part. As clarified throughout this point-by-point rebuttal, cats were classified as FIV- or FeLV-positive according to their serological status obtained during admission or hospitalization, regardless of whether the retroviral infection itself or another concomitant disorder was the primary reason for hospitalization. Some cats were admitted due to retroviral infection itself, but others were admitted for different clinical conditions and were also found to be seropositive.
To address this concern, we have revised the manuscript throughout to replace ambiguous expressions such as “being hospitalized with FIV/FeLV” with the more accurate terms “FIV seropositivity” or “FeLV seropositivity.” These revisions were made in the Results (Section 3.1, 3.2 and 3.3, page 6,7,8 and 10 lines 254, 262, 269, 272, 276, 301 and 305), Discussion (Sections 4.1 and 4.2, pages 11, 12, 13 and 14, lines 318–320, 324, 329, 331, 341, 351, 360, 366, 373, 379, 386-387, 396, 398, 401, 410, 418, 420, 422, 434, 436), and Conclusions (Section 5, page 15, line 495). Additionally, in the Methods (Section 2.2, page 3, lines 134-136).
Comments 14: Line 438 – 446: There are some typos in this paragraph.
Reply: We thank the reviewer for noting this. The typos have been corrected (Page 14, lines 474–479).
Comments 15: Line 468: I don’t understand why the model needs to be developed at each individual BICU. The variables you collected would be part of a standard cat history. What would be different in the populations that would make this non-generalizable.
Reply: Thank you for the comment. The variables included in our models are indeed part of a standard feline history and, in principle, can be applied more broadly. However, our dataset was derived from a referral population hospitalized in a BICU under suspicion or confirmation of infectious disease, which differs from the general cat population. This setting likely influences both disease prevalence and the relative weight of predictors. To clarify this point, we have emphasized that while the variables are general, external validation in independent populations is required for the findings to be generalizable to other popultions. This clarification can be found in the Conclusions section (Page 15, Lines 501–509): “Follow-up work should be conducted towards in order to perform model validation with data external to the unit, to improve the models and their sensitivity/specificity, and to develop a computer application based on the constructed models for the respective deployment into the unit. Although the variables used are part of standard feline history and examination, the case-mix in our study was shaped by the specific referral profile of the BICU, with a higher prevalence of infectious disease suspicions. This context may influence the relative predictive weight of individual factors. Therefore, we recommend that similar modeling exercises be undertaken in other units, allowing validation and, if necessary, adaptation of the models to the characteristics of their own patient populations.”
Comments 16: I didn’t see HEV-FMB, which I’m guessing refers to a specific BICU?
Reply: Thank you for spotting this typo. The correct abbreviation is VTH-FMV: Veterinary Teaching Hospital of the Faculty of Veterinary Medicine, which has now been included in the abbreviations list and corrected throughout the text.
Comments 17: Minor, but ref 14 is missing the journal name.
Reply: We thank the reviewer for spotting this incomplete reference. We have now included the journal name in the respective reference.
Reviewer 2 Report
Comments and Suggestions for Authors
Please see the attached file.

Author Response
Comments 1: The title is quite long and could be improved, here's a suggestion: “Statistical triage model for feline infectious diseases in a veterinary isolation unit: The case of feline immunodeficiency and leukemia viruses”
Reply: Thank you for the suggestion. We now have modified the title accordingly (Page 1, lines 1-4).
Comments 2: Abstract, line 38: the final number of cats included in the evaluation was lower, please specify.
Reply: Thank you for this suggestion. We have now specified the eligible number of cats analysed (n=640; Page 1, lines 37–38).
Comments 3: Abstract, lines 44-45: please describe here the tests done for the diagnosis of both infections.
Reply: Thank you for the suggestion. We have revised the abstract as follows (Page 1, lines 40–42): “Our model training was based on data from 134 FIV-seropositive cats, 126 FeLV-seropositive, and 504 confirmed non-cases (i.e., controls) of these infections diagnosed by rapid immunomigration assays and ELISA.”
Comments 4: Introduction, lines 77-78: “, lymphoma, and secondary infections” please change with “comorbidities and coinfections”.
Reply: Thank you for the suggestion. We have revised the sentence accordingly (Page 2, lines 73–74).
Comments 5: Introductions, lines 81-85: the diagnosis of FeLV as you explain in the discussion is more complex, please mention that here as well.
Reply: Thank you for the suggestion. We have now explained the complexity of FeLV diagnosis in the Introduction (Page 2, lines 82–85) reads: "However, diagnosing FeLV infection is challenging because the virus can establish different outcomes—progressive, regressive, abortive, or focal—that are not always reliably distinguished by point-of-care tests, requiring molecular confirmation in some cases [12].”
Comments 6: “Participants and Data Collection” paragraph: it is not explained whether the FIV- and FeLV-positive cats were sick patients or not, and what their clinical signs were. It is also not explained whether the control group cats were healthy patients (according to clinical and laboratory evaluation) or simply FIV- and FeLV-negative, as this is an important limitation of the study. As you explain in the discussion paragraph FIV-infected cats can remain asymptomatic or mildly symptomatic for an extended period, and the progression of the disease varies among individuals, so in the asymptomatic phase you may not notice hematological changes. As well as diagnosis of progressive or regressive FeLV infection require different diagnostic approaches, therefore the use of the ELISA test alone may have affected the diagnosis of some subjects.
Reply: We thank the reviewer for raising this point; similar point was raised by another reviewer. We have now clarified that cats admitted to the BICU by referral due to a suspicion or confirmed infectious disease and thus included both clinically sick animals and those with milder or nonspecific signs. For the purpose of this study, cats were classified as FIV- or FeLV-positive according to their serological test results obtained during admission or hospitalization, regardless of whether retroviral infection itself or another concomitant disorder was the primary reason for hospitalization. With respect to the control group, these cats were defined as retrovirus-negative according to diagnostic testing but were not necessarily clinically healthy. Furthermore, as discussed in Sections 4.1 and 4.2, FIV-seropositive cats may remain asymptomatic for extended periods, and regressive FeLV infections cannot be reliably diagnosed by ELISA or immunomigration tests alone. We now acknowledge this diagnostic limitation explicitly in the Methods. The revised text (Pages 4 and 5, lines 134–144) reads: “The population of FIV and FeLV-positive cats admitted to the BICU included both clinically sick animals and those presenting with milder or nonspecific signs, reflecting the routine case mix of suspected infectious disease patients. For the purpose of this study, cats were classified as FIV or FeLV-positive according to their serological test results obtained at admission or during hospitalization, regardless of whether retroviral infection itself or another concomitant disorder was the primary reason for hospitalization. Control group cats were defined as retrovirus-negative according to diagnostic testing [17,18] but were not necessarily clinically healthy. This approach reflects the clinical reality of hospital admissions at BICU. Moreover, given that FIV-infected cats may remain asymptomatic for extended periods and regressive FeLV infections may not be detected by ELISA or immunomigration tests alone, some degree of misclassification cannot be excluded”
Comments 7: ”Sample testing” paragraph: can you specify at the beginning of the paragraph that different diagnostic tests were used? Please also specify how many animals were evaluated with the various tests.
Reply: We thank the reviewer for this suggestion. We have revised the beginning of the “Sample Testing” paragraph to explicitly state that different diagnostic tests were used during the study period and specified the number of animals evaluated with each test (Page 4, lines 156–168): Viracheck® (FeLV and FIV ELISA Test, n=606); Vetline® (FIV Antibody Test; n=124); WITNESS® (FeLV-FIV Test, n=93); and PCR (n=15)”.
Comments 8: Lines 159-161: “In some cases, additional laboratory ELISA testing was performed to confirm or clarify point-of-care results. However, this test was not applied routinely, as well as the application of confirmatory testing, such as PCR or Western blot.” Which results did you use in this case? The confirmed ones or the starting result? Please specify.
Reply: Thank you for this question. In routine practice, ELISA was the standard diagnostic test. Point-of-care immunomigration assays were often used during consultations for rapid screening, and ELISA was applied to confirm or clarify uncertain cases. In rare situations where ELISA results were inconclusive, PCR or Western blot were used for confirmation. For this study, the final confirmed result was always used in the analysis. We have clarified the diagnostic workflow in the “Sample Testing” section now reads: “In routine diagnostic workflows, Enzyme-Linked Immunosorbent Assays (ELISA)…” (Page 4, lines 152–153); “… were also available and were typically used in consultations for rapid screening, given their ease of use and short turnaround. In cases where these results required confirmation or clarification, ELISA testing was subsequently performed.” (Page 4, lines 163–165); “In some cases, additional confirmatory testing with PCR (n=15) or Western blot was performed to confirm or clarify ELISA results. However, these tests were not applied routinely. For this study, when confirmatory testing was applied, the final confirmed result was considered for analysis.” (Page 4, lines 167–170).
Comments 9: Line 211: I assume the CBC was the most readily available lab data, but why only assess the hematocrit and white blood cell count? Please explain this in the introduction and discussion.
Reply: We thank the reviewer for this question. Hematocrit and total leukocyte count were indeed selected, because these biomarkers are the most consistently documented CBC parameters in our clinical records. Other hematological variables were available but they had higher rates of missing data (>10%) and were excluded from the analysis for simplicity. We have clarified this point in the Materials and Methods (page 5, lines 220-222) and now reads: “We decided to exclude from this analysis all variables included in the database with more than 10% missing data, as done elsewhere [26]; this applied specifically to origin, vaccination status and other CBC parameters.
We also acknowledged the following in the Discussion (page 12, lines 362-364) “Other hematological values were excluded as they had more than 10% missing data. We applied a 10% cutoff for simplicity, opting for exclusion rather than imputation.”
Comments 10: Line 225: please remove the double “using”
Reply: Thank you for spotting this doubling. We have corrected accordingly.
Comments 11: Lines 252-254: “There were evidence for significant associations between FeLV infection and age group (p <0.001), hematocrit classification (p < 0.001), breed (p = 0.037), and the presence of concomitant disorders (p =0.046).” Please specify which age group, which hematocrit, which breed.
Reply: We thank the reviewer for this helpful comment. We have therefore revised the Results section to indicate which groups, based on the univariate logistic regression estimates and p-values, seemed to contribute to the significant associations. The revised text (Page 6, lines 255–261 and 263-267) now reads:
“There were statistically significant associations between FIV infection and age group, sex-neuter status, lifestyle, and the presence of concomitant disorders. Adult (estimate = 1.636, p < 0.001) and senior cats (estimate = 2.149, p < 0.001), intact males (estimate = 0.757, p = 0.031), and cats with outdoor access (estimate = 2.324, p < 0.001) seemed to positively contribute to these associations, as did the presence of concomitant disorders (estimate = 0.777, p = 0.001) (Table 2). Other variables were not significantly associated with FIV status.”
“There was evidence for significant associations between FeLV infection and breed, age group, hematocrit, and concomitant disorders. Mixed-breed cats (estimate = 1.143, p = 0.031), cats aged 2–9 years (estimate = 0.616, p = 0.016), low hematocrit values (estimate = 0.886, p = 0.017), and the presence of concomitant disorders (estimate = 0.497, p = 0.036) seemed to contribute to these associations (Table 3).”
Comments 12: Line 298: Please also summarize the results obtained for FIV infection at the beginning of the paragraph.
Reply: We thank the reviewer for the suggestion. We have now begun the paragraph (Page 11, lines 318-321) as follows:
“In our analysis of FIV seropositivity, outdoor access, presence of concomitant diseases and older age groups (≥2 years) were significantly associated with increased likelihood of FIV seropositivity, while no statistically significant associations were observed for sex-neuter status, breed, hematocrit, or leukocyte count.”
Comments 13: Line 347: Please also summarize the results obtained for FeLV infection at the beginning of the paragraph.
Reply: Thank you for the suggestion. We have now begun the paragraph (page 12, lines 373-377) as follows: “The multivariable analysis of FeLV seropositivity revealed significant associations with mixed-breed status, the presence of concomitant diseases, and reduced hematocrit values. By contrast, sex-neuter status, outdoor access, multi-cat households, and leukocyte counts were not independently associated. The effect of age is significant in the univariate analysis, but this did not persist in the final model.”
Comments 14: Line 439: please correct with “In fact”
Reply: Thank you for noticing this typo. We have revised the text accordingly (Page 14, line 475).
Comments 15: Line 440: please remove the dot after “period”
Reply: Thank you for noticing this typo. We have revised the text accordingly (Page 14, line 476).
Comments 16: Line 443: please remove the double dot
Reply: Thank you for noticing this typo. We have revised the text accordingly (Page 14, line 479).
Reviewer 3 Report
Comments and Suggestions for Authors
Although the author believes that a preliminary triage model for FeLV and FIV infections has been established, if clinical practice were to follow this triage model for triage, the sensitivity and specificity would not meet the standards. Many other factors, especially specific diagnostic methods, need to be combined. Moreover, the so-called triage model has not been verified. This is equivalent to no triage model being established; it is merely a statistical calculation of previous case data.
Author Response
Comment: Although the author believes that a preliminary triage model for FeLV and FIV infections has been established, if clinical practice were to follow this triage model for triage, the sensitivity and specificity would not meet the standards. Many other factors, especially specific diagnostic methods, need to be combined. Moreover, the so-called triage model has not been verified. This is equivalent to no triage model being established; it is merely a statistical calculation of previous case data.
Reply: We thank the reviewer for this comment. We agree that the sensitivity and specificity of our models did not meet the standards required for decision-making in isolation. However, this work should be seen as an initial iteration in the process of developing more informative triage models. In fact, a good analogy for our study is to consider it as phase I clinical trial. As such, the models developed set the foundation for future model refinement and validation similarly to what happens in phases II, III and IV clinical trials.
We acknowledge the reviewer’s point that many other factors are needed to strengthen the models. In fact, we emphasized in the Discussion and Limitations that our retrospective dataset was restricted to routinely collected variables and did not include more specific diagnostic markers or confirmatory molecular tests. The statistical scope is inherent to the proof-of-principle nature of our study, and we agree that future iterations of triage models should integrate broader clinical, laboratory, and diagnostic information.
Regarding the observation that our work is merely a statistical calculation of previous case data, we note that, as suggested by another reviewer, we revised the title to emphasize precisely this statistical approach.
All these points were discussed in lines 1-4 (title), lines 460-465 (additional data), lines 466-475 (combination of different diagnostics), lines 485-490 (model validation), lines 501-509 (follow-up work).
Round 2
Reviewer 1 Report
Comments and Suggestions for Authors
Thank you for your careful consideration of the comments. The Methods section is much clearer and the limitations are better acknowledged. All of my comments have been addressed.
Author Response
We thank the reviewer for the time and effort in helping us to improve our manuscript
Reviewer 2 Report
Comments and Suggestions for Authors
The manuscript may be accepted in this modified form.
Author Response

(The authors gave the same response as above.)

Reviewer 3 Report
Comments and Suggestions for Authors
According to the model, the area under the receiver operating characteristic curve for FIV-infected subjects was 0.70, and for FeLV-infected subjects it was 0.69. The estimated sensitivity and specificity values were both higher than 65%. However, they were far from reaching the level of sensitivity and specificity exceeding 90%. Through rapid immunochromatographic assay and ELISA testing, the sensitivity for diagnosing these infections was 92.6% - 95.5%, and the specificity was 93.4% - 99.8%. Therefore, relying solely on the model for triage only plays a partial role in alleviating the pressure on the hospital. It is necessary to consider combining rapid immunochromatographic assay and ELISA testing for diagnosis to maximize the effect of alleviating hospital pressure, but the time before hospitalization will be extended. So, relying solely on the so-called triage model to alleviate hospital pressure is of little significance. The research results indicate that this triage model is not feasible unless the title and content of the article are significantly modified. That is, to reduce the workload of testing, initially classify the cats in the outpatient department based on clinical data, and then determine the cats that need to be hospitalized due to FIV or FeLV infection according to the results of rapid immunochromatographic assay and ELISA testing.
Author Response
We thank the reviewer for this observation. We agree that point-of-care tests and ELISA provide higher sensitivity and specificity than our statistical models. However, such testing may not always be immediately accepted, as costs (around 80 € per cat) are directly covered by the owners and can represent a significant burden. In this context, our aim was not to replace diagnostic tests, but to explore whether routinely collected clinical data could serve as a no-cost, complementary triage aid. Such a model may help clinicians prioritize animals with higher probability of retroviral infection and strengthen discussions with owners regarding the need for confirmatory testing.
To clarify this point, we have revised the manuscript in two sections:
• Materials and Methods (page 3, lines 115–118 and 121–126): We now explain the context of diagnostic costs and explicitly state how a model based on clinical data could support prioritization and communication with owners.
• Conclusions (page 15, lines 504–509): We revised the text to emphasize that the models are preliminary, no-cost decision-support tools, not substitutes for point-of-care or ELISA assays.
Above all, these models are not intended to replace diagnostic tests; rather, they represent a first attempt to understand what level of performance could be achieved using this type of routinely collected data.
Regarding the title, we acknowledge that the initial version of the manuscript may have given the impression that a fully feasible triage model had been established. To avoid this misinterpretation and following the suggestion of another reviewer in the previous round, we have already modified the title to emphasize the statistical and preliminary nature of the approach.